# Sparse Graphical Memory for Robust Planning

**Scott Emmons**[*]
Berkeley AI Research

**Ajay Jain**[*]
Berkeley AI Research

**Michael Laskin**[*]
Berkeley AI Research

**Thanard Kurutach**
Berkeley AI Research

**Pieter Abbeel**
Berkeley AI Research

**Deepak Pathak**
Carnegie Mellon University

## Abstract

To operate effectively in the real world, agents should be able to act from high-dimensional raw sensory input such as images and achieve diverse goals across long time-horizons. Current deep reinforcement and imitation learning methods can learn directly from high-dimensional inputs but do not scale well to long-horizon tasks. In contrast, classical graphical methods like A* search are able to solve long-horizon tasks, but assume that the state space is abstracted away from raw sensory input. Recent works have attempted to combine the strengths of deep learning and classical planning; however, dominant methods in this domain are still quite brittle and scale poorly with the size of the environment. We introduce Sparse Graphical Memory (SGM), a new data structure that stores states and feasible transitions in a sparse memory. SGM aggregates states according to a novel two-way consistency objective, adapting classic state aggregation criteria to goal-conditioned RL: two states are redundant when they are interchangeable both as goals and as starting states. Theoretically, we prove that merging nodes according to two-way consistency leads to an increase in shortest path lengths that scales only linearly with the merging threshold. Experimentally, we show that SGM significantly outperforms current state of the art methods on long horizon, sparse-reward visual navigation tasks. Project video and code are available at https://mishalaskin.github.io/sgm/.

## 1 Introduction

Learning-driven approaches to control, like imitation learning and reinforcement learning, have been quite successful in both training agents to act from raw, high-dimensional input [34] as well as to reach multiple goals by conditioning on them [1, 35]. However, this success has been limited to short horizon scenarios, and scaling these methods to distant goals remains extremely challenging. On the other hand, classical planning algorithms have enjoyed great success in long-horizon tasks with distant goals by reduction to graph search [18, 25]. For instance, A* was successfully used to control *Shakey the robot* for real-world navigation over five decades ago [7]. Unfortunately, the graph nodes on which these search algorithms operate

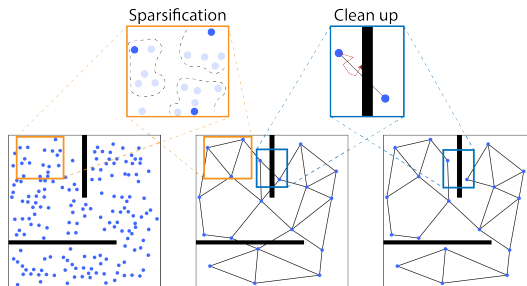

Figure 1: Illustration of Sparse Graphical Memory (SGM). New states are either merged with existing graph nodes according to our two-way consistency criterion, or a new node is generated. After graph construction, incorrect edges representing infeasible transitions are corrected.

[*]Equal contribution. Author order determined randomly. {emmons, ajayj, mlaskin}@berkeley.edu

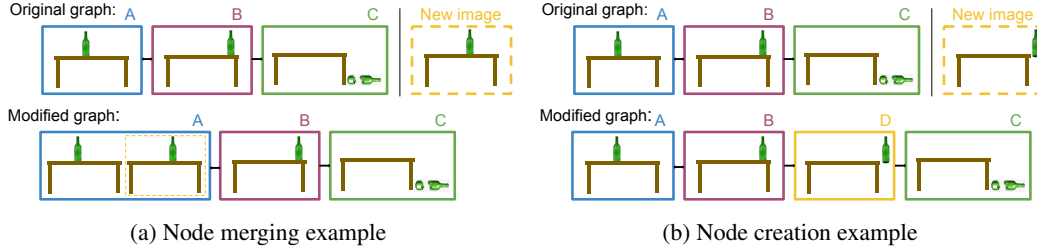

(a) Node merging example           (b) Node creation example

Figure 2: Examples where two-way consistency (TWC) merges nodes (left) and creates a new node (right). Consider a directed graph with three nodes connected as $A \leftrightarrow B \to C$. Given a new image in the dashed yellow box, should it be merged into an existing node or a new node be created? (a) On the left, we can merge the new image with node A safely. (b) On the right, the new image D contains a bottle which is about to fall off the table edge. While B is perceptually similar to the new image, the agent cannot move the bottle from D to A, but it can move from B to A. As we cannot transition to neighbors in the same manner, our TWC criterion is not satisfied and a new node is created.

is abstracted away from raw sensory data via domain-specific priors, and planning over these nodes assumes access to well-defined edges as well as a perfect controller to move between nodes. Hence, these planning methods struggle when applied to agents operating directly from high-dimensional, raw-sensory images [32].

How can we have best of both worlds, *i.e.*, combine the long-horizon ability of classic graph-based planning with the flexibility of modern, parametric, learning-driven control? One way is to build a graph out of an agent's experience in the environment by constructing a node for every state and use a learning-based controller (whether RL or imitation) to move between those nodes. Some recent work has investigated this combination in the context of navigation [9, 41]; however, these graphs grow quadratically in terms of edges and quickly become unscalable beyond small mazes [9]. This strategy either leads to extremely brittle plans because such large graphs contain many errors (infeasible transitions represented by edges), or relies on human demonstrations for bootstrapping [41].

In this work, we propose to address challenges in combining the classical and modern paradigms by dynamically sparsifying the graph as the agent collects more experience in the environment to build what we call *Sparse Graphical Memory* (SGM), illustrated in Figure 1. In fact, building a sparse memory of key events has long been argued by neuroscientists to be fundamental to animal cognition. The idea of building cognitive topological maps was first demonstrated in rats by seminal work of [47]. The key aspect that makes building and reasoning over these maps feasible in the ever-changing, dynamic real world is the sparse structure enforced by landmark-based embedding [12, 14, 48]. Yet, in artificial agents, automatic discovery of sparse landmark nodes remains a key challenge.

One way to discover a sparse graph structure is to dynamically merge similar nodes. But how does one obtain a similarity measure? This is a subtle but central piece of the puzzle. States that look similar in the observation space may be far apart in the action space, and vice-versa. Consider the example in Figure 2, where the graph already contains 3 nodes $\{A, B, C\}$. In 2a, similar looking nodes can be merged safely. While in 2b, although the new node $D$ is visually similar to $B$, but merging with $B$ would imply that the bottle can be saved from breaking. Therefore, a purely visual comparison of the scenes cannot serve as a viable metric. We propose to use an asymmetric distance function between nodes and employ *two-way consistency* (TWC) as the similarity measure for merging nodes dynamically. The basic idea is that two nodes are similar if they both can be reached with a similar number of steps from all their neighbors as well as if all their neighbors can be reached from both of them with similar effort. For our conceptual example, it is not possible to go back from the falling-bottle to the standing-bottle, and hence the two-way consistency does not align for scene $B$ and the new state. Despite similar visual appearance, they will not be merged. We derive two-way consistency as an extension of prior Q-function based aggregation criteria to goal-conditioned tasks, and we prove that the sparse graphs that result from TWC preserve (up to an error factor that scales linearly with the merging threshold) the quality of the original dense graphs.

We evaluate the success of our method, SGM, in a variety of navigation environments. First, we observe in Table 1 that SGM has a significantly higher success rate than previous methods, *on average increasing the success rate by 2.1x* across the environments tested. As our ablation experiments demonstrate, SGM's success is due in large part to its sparse structure that enables efficient correction

of distance metric errors. In addition, we see that the performance gains of SGM hold across a range of environment difficulties from a simple point maze to complex visual environments like ViZDoom and SafetyGym. Finally, compared to prior methods, planning with our proposed sparse memory can lead to nearly an order of magnitude increase in speed (see Appendix E).

## 2  Related work

Planning is a classic problem in artificial intelligence. In the context of robotics, RRTs [25] use sampling to construct a tree for path planning in configuration space, and SLAM jointly localizes the agent and learns a map of the environment for navigation [2, 8]. Given an abstract, graphical representation of an environment, Dijkstra's Algorithm [6] generalizes breadth-first search to efficiently find shortest paths in weighted graphs, and the use of a heuristic function to estimate distances, as done in A* [18], can improve computational efficiency.

Beyond graph-based planning, there are various parametric approaches to planning. Perhaps the most popular planning framework is model predictive control (MPC) [13]. In MPC, a dynamics model, either learned or known, is used to search for paths over future time steps. To search for paths, planners solve an optimization problem that aims to minimize cost or, equivalently, maximize reward. Many such optimization methods exist, including forward shooting, cross-entropy, collocation, and policy methods [17, 40]. The resulting agent can either be open-loop and just follow its initial plan, or it can be closed-loop and replan at each step.

Aside from MPC, a variety of reinforcement learning algorithms, such as policy optimization and Q-learning, learn a policy without an explicit dynamics model [27, 33, 43, 44]. In addition to learning a single policy for a fixed goal, some methods aim to learn hierarchical policies to decompose complex tasks [22, 36, 42], and other methods aim to learn goal-conditioned policies able to reaching arbitrary goals. Parametric in nature, these model-free approaches are highly flexible, but, as does MPC with a learned dynamics model, they struggle to plan over long time horizons due to accumulation of error.

Recent work combines these graph-based and parametric planning approaches by using past observations for graph nodes and a learned distance metric for graph edges. Variations of this approach include Search on the Replay Buffer [9], which makes no attempt to sparsify graph nodes; Semi-Parametric Topological Memory [41], which assumes a demonstration to bootstrap the graph; Mapping State Space Using Landmarks for Universal Goal Reaching [20], which subsamples the policy's past training observations to choose graph nodes; and Composable Planning with Attributes [50], which stores abstracted attributes on each node in the graph. Hallucinative Topological Memory (HTM) [28] uses a contrastive energy model to construct more accurate edges, and [45] use dynamic programming for planning with a learned graph.

In contrast to the methods listed above, the defining feature of our work is a two-way consistency check to induce sparsity. Previous work either stores the entire replay buffer in a graph, limiting scalability as the graph grows quadratically in the number of nodes, or it subsamples the replay buffer without considering graph structure. Moreover, Neural Topological SLAM [4] assumes access to a 360°camera and a pose sensor whereas we do not. In [5], the nodes in the graph are manually specified by humans whereas we automatically abstract nodes from the data. In contrast to our method, the work in [31] has no theoretical guarantees, requires trajectories rather than unordered observations, and uses human demonstrations.

Prior work has proposed MDP state aggregation criteria such as bisimulation metrics that compare the dynamics at pairs of states [11, 15] and Utile distiction [30] that compares value functions. Because bisimulation is a strict criterion [26] that would not result in meaningful sparsity in the memory, our proposed two-way consistency criteria adapts approximate value function irrelevance [21] to goal-conditioned settings and high-dimensional observations.

## 3  Preliminaries

We consider long-horizon, goal-conditioned tasks. At test time, an agent is provided with its starting state $s_{\text{start}}$ and a goal state $s_{\text{goal}}$, and the agent seeks to reach the goal state via a sequential decision making process. Many visual tasks can be defined by a goal state such as an image of a goal location

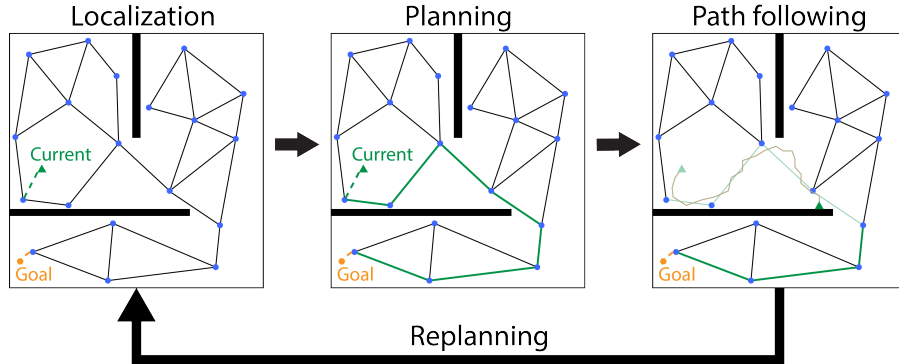

Figure 3: Execution using a graphical memory. In localization, the agent finds the closest node using discrepancies in the asymmetric distance function. In planning, the agent uses Dijkstra's algorithm to find the shortest path. (For simplicity, this illustration omits the direction of edges.) In path following, the agent may divergence from the waypoints or experience a transition failure. The agent then needs to correct the memory, relocalize, and replan.

for navigation. Our task is an instance of undiscounted finite-horizon goal-conditioned RL in which the agent receives a -1 living reward until it reaches a given goal.

To reach distant goals, we use a semi-parametric, hierarchical agent that models feasible transitions with a nonparametric graph (*i.e.* graphical memory) to guide a parametric low-level controller.

Nodes in the graphical memory are prior states encountered by the agent, connected through a learned distance metric. Once the graphical memory is constructed, the agent can plan using a graph search method such as Dijkstra's algorithm [6] and execute plans with a low-level controller learned with reinforcement learning or imitation learning [9, 41]; see Figure 3. Therefore, given a final goal, the high-level planner acts as a policy that provides waypoints, nodes along the shortest path, to the low-level controller, and updates the plan as the low-level agent moves between waypoints:

$$\pi^{hl} \text{ uses } Q^{hl}(s, a = \omega | g) \text{ to select a graph waypoint } \omega \text{ to reach the final goal } g$$
$$\pi^{ll} \text{ uses } Q^{ll}(s, a | \omega) \text{ to take an environment step } a \text{ to reach the waypoint } \omega.$$

As the agent receives an undiscounted -1 living reward, the optimal low-level value function measures the number of steps to reach a waypoint, *i.e.* the distance $d(s, \omega) = -\max_a Q^{ll}(s, a | \omega)$.

The main challenge with the above semi-parametric formalism is the detrimental effect of errors in the graphical memory. Since nodes are connected by an approximate metric, the number of errors in the graph grows as $O(|\mathcal{V}|^2)$ where $|\mathcal{V}|$ is the number of graph nodes. Moreover, in order for graphical memory methods to scale to larger environments, it is infeasible to store every state encountered in the graph as done in prior work [9, 41]. For the above reasons, node sparsity is a desirable feature for any graphical memory method. In this work, we seek to answer the following research question: *given a dense graphical memory, what is the optimal algorithm for transforming it into a sparse one?*

## 4  Sparse Graphical Memory

The number of errors in graphical memory can be minimized by removing redundant states, as each state in the memory can introduce several infeasible transitions through its incident edges. Any aggregation algorithm must answer a key question: *when can we aggregate states?*

Sparse Graphical Memory is constructed from a buffer of exploration trajectories by retaining the minimal set of states that fail our *two-way consistency* aggregation criterion. In Sec. 4.1, we introduce this two-way consistency criterion, which extends prior work to the goal-conditioned RL setting. We then prove that as long as a sparsified graph satisfies two-way consistency, plans in the sparsified graph are close to the optimal plan in the original graph. In Sec. 4.2, we design an online algorithm for checking two-way consistency by selecting the approximate minimal set from the replay buffer. Finally, in Sec. 4.3, we outline a cleanup procedure for removing remaining infeasible transitions.

## 4.1 State aggregation via two-way consistency

As a state aggregation criterion, prior work argues for value irrelevance by aggregating states with similar Q-functions [21, 26, 30]. If the optimal Q-functions are the same at two states, the optimal policy will select the same action at either state.

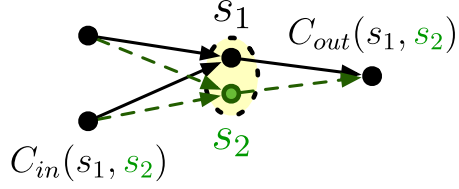

However, in the goal-conditioned setting, we face a challenge: the states in the memory define the action space of the high-level policy, *i.e.* the waypoints selected by the graph planner. Aggregating states changes the action space by removing possible waypoints.

Figure 4: SGM uses a two-way consistency criterion to find redundant pairs of states in the replay buffer.

In order to extend value irrelevance to the goal-conditioned setting, we propose two-way consistency (TWC). Under two-way consistency, two states are redundant if they are both interchangeable as goals and interchangeable as starting states according to the goal-conditioned value function.

The graph planner acts as a nonparametric high-level policy, selecting a waypoint $\omega$ for the low-level controller as an intermediate goal. Then, the action space is given by candidate waypoints. Formally, using the optimal value function of the high-level policy, states $s_1$ and $s_2$ are interchangeable as starting states if

$$\max_\omega \left| Q^{hl}(s_1, a = \omega|g) - Q^{hl}(s_2, a = \omega|g) \right| \leq \tau_a. \tag{1}$$

Then, as $\tau_a \to 0$, the high-level policy will behave the same in both states. Equation (1) can be seen as verifying that the sparsification is a $\tau_a$-approximate $Q$-irrelevant abstraction [21] given a fixed set of waypoints as actions. Further, we say that states $s_1$ and $s_2$ are interchangeable as waypoints for the purposes of planning if

$$\max_{s_0} \left| Q^{hl}(s_0, a = s_1|g) - Q^{hl}(s_0, a = s_2|g) \right| \leq \tau_a. \tag{2}$$

As a consequence, eliminating $s_2$ from the memory and thus from the action space will not incur a loss in value at any starting state as the policy can select $s_1$ instead. Together, (1) and (2) form the two directions of our two-way consistency criterion in the high-level value space.

While we have motivated two-way consistency in terms of $Q^{hl}$ as $\tau_a \to 0$, it furthermore holds that two-way consistency gives a meaningful error bound for *any asymmetric distance function* $d(\cdot, \cdot)$ satisfying $C_{out}(s_1, s_2) \leq \tau_a$ and $C_{in}(s_1, s_2) \leq \tau_a$ for any finite $\tau_a$ where

$$C_{out}(s_1, s_2) := \max_\omega |d(s_1, \omega) - d(s_2, \omega)| \text{ and } C_{in}(s_1, s_2) := \max_{s_0} |d(s_0, s_1) - d(s_0, s_2)|. \tag{3}$$

The following theorem shows that aggregating states according to two-way consistency of $d(\cdot, \cdot)$ incurs a bounded increase in path length that scales linearly with $\tau_a$. Furthermore, given an error bound on the distance function $d(\cdot, \cdot)$, it provides an error bound on distances in the sparsified graph.

**Theorem 1.** *Let $\mathcal{G}_\mathcal{B}$ be a graph with all states from a replay buffer $\mathcal{B}$ and weights from a distance function $d(\cdot, \cdot)$. Suppose $\mathcal{G}_{TWC}$ is a graph formed by aggregating nodes in $\mathcal{G}_\mathcal{B}$ according to two-way consistency $C_{out}$ and $C_{in} \leq \tau_\alpha$. For any shortest path $P_{TWC}$ in $G_{TWC}$, consider the corresponding shortest path $P_\mathcal{B}$ in $G_\mathcal{B}$ that connects the same start and goal nodes. Suppose $\mathcal{P}_\mathcal{B}$ has $k$ edges. Then:*

*(i) The weighted path length of $P_{TWC}$ minus the weighted path length of $P_\mathcal{B}$ is no more than $2k\tau_\alpha$.*

*(ii) Furthermore, if $d(\cdot, \cdot)$ has error at most $\epsilon$, the weighted path length of $P_{TWC}$ is within $k\epsilon + 2k\tau_\alpha$ the true weighted distance along $\mathcal{P}_\mathcal{B}$.*

Theorem 1 is proved in Appendix D. We emphasize that it makes no assumptions on the distance function, but relies on both directions of our criterion. This distance can be derived from the value function by negation or learned in a supervised manner, depending on the application. According to Theorem 1, we can aggregate pairs of states that satisfy two-way consistency without leading to significantly longer plans (only a constant increase). Since two-way consistency is approximate, at the same time, it also leads to more sparsity than restrictive criteria like bisimilarity.

In contrast to two-way consistency, a naïve strategy like random subsampling of the replay buffer does not provide a guarantee without further assumptions on the exploration data such as an even

---

**Algorithm 1** `BuildSparseGraph`

---

1: **Input:** replay buffer $\mathcal{B}$, distance function $d$
2: **Output:** sparse graph $\mathcal{G} = (\mathcal{V}, \mathcal{E}, \mathcal{W})$
3: Initialize empty vertex set $\mathcal{V} = \emptyset$
4: **for** $\hat{s} \in \mathcal{B}$, each state in the replay buffer, **do**
5:     **if** state is novel according to TWC, *i.e.* $C_{in}(s, \hat{s}) \leq \tau_a, C_{out}(s, \hat{s}) \leq \tau_a \; \forall s \in \mathcal{V}$ **then**
6:         add the state $\hat{s}$ to the graph $\mathcal{G}$:
7:         $\mathcal{V} = \mathcal{V} \cup \{\hat{s}\}$
8:         $\mathcal{E} = \mathcal{E} \cup \{(s, \hat{s}) : s \in \mathcal{V}, d(s, \hat{s}) \leq \text{MAXDIST}\} \cup \{(\hat{s}, s) : s \in \mathcal{V}, d(\hat{s}, s) \leq \text{MAXDIST}\}$
9:     **end if**
10: **end for**
11: assign weights $\mathcal{W}(s_i, s_j) = d(s_i, s_j) \;\; \forall (s_i, s_j) \in \mathcal{E}$
12: filter transition set $\mathcal{E}$ to $k$ nearest neighbors
13: **return** $\mathcal{G} = (\mathcal{V}, \mathcal{E}, \mathcal{W})$

---

coverage of the state space. For example, when randomly subsampling the buffer, a rarely visited bottleneck state will be dropped with the same probability as states in frequently visited regions. As multiple states can be covered by the same state, two-way consistency is a prioritized sparsification of the replay buffer that seeks to cover the environment regardless of sampling density.

### 4.2 Constructing the graph from a replay buffer

State aggregation can be seen as an instance of set cover, a classic subset selection problem. In our instantiation, we select a subset of the states in the replay buffer to store in the agent's memory so that all states in the replay buffer are "covered". Coverage is determined according to the state aggregation criterion, two-way consistency (3). Unfortunately, set cover is a combinatorial optimization problem that is NP-hard to solve exactly, or even to approximate to a constant factor [10, 29, 39].

Motivated by the difficulty of the set cover problem, we propose an online, greedy algorithm for replay buffer subset selection. The graphical memory is built via a single pass through a replay buffer of experience according to Algorithm 1, which requires quadratically many evaluations of the TWC criterion. In particular, a state is only recorded if it is novel according to the two-way consistency criterion. Once a state is added to the graph, we create incoming and outgoing edges, and set the edge weight to the distance.

### 4.3 Graphical memory cleanup

Although TWC produces a compact graph with substantially fewer errors than the original graphical memory, faulty edges may still remain. If even one infeasible transition between distant states remains as an edge in the graph, the planner will exploit it, and the agent will not be able to carry out the plan [9]. Therefore, eliminating faulty edges is key for robustness.

We bound the number of errors in the memory by filtering edges, limiting nodes to their $k$ nearest successors. After filtration, the worst-case number of untraversable edges grows only *linearly* in the sparsified node count, not quadratically. This is a simple, inexpensive procedure that we experimentally found removes many of the infeasible transitions.

Finally, we propose a novel self-supervised error removal technique – *cleanup*. During test-time execution, a random goal $g$ is sampled and the planner provides a set of waypoints $w$ for the low-level agent. When the low-level agent is unable to reach a consecutive waypoint $w_i \rightarrow w_{i+1}$, it removes the edge between $(w_i, w_{i+1})$ and re-plans its trajectory. This procedure is efficient for a TWC sparsified graph since $|\mathcal{G}_{\mathcal{TWC}}| \ll |\mathcal{G}_{\mathcal{B}}|$.

## 5 Experimental Setup

We evaluate SGM under two high-level learning frameworks: reinforcement learning (RL), and self-supervised learning (SSL). As a general data structure, SGM can be paired with any learned

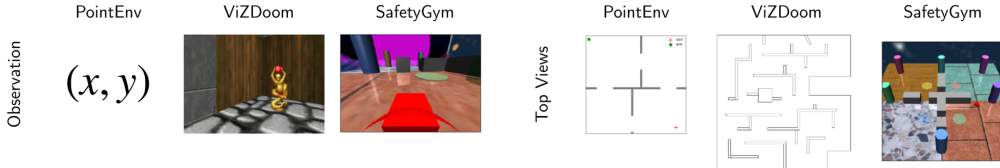

Figure 5: The three environments used for testing SGM. PointEnv is a small maze with coordinate observations. We increase its difficulty by thinning the walls. ViZDoom is a large environment, which can take up to 5 minutes and 5k steps to traverse entirely. ViZDoom actions are discrete and observations are first-person camera views. SafetyGym is another large environment with first-person view observations, and supports continuous actions.

| TECHNIQUE | SUCCESS RATE | CLEANUP STEPS | OBSERVATION | ENV |
|---|---|---|---|---|
| SoRB | $28.0 \pm 6.3\%$ | 400K | PROPRIO | POINTENV |
| SoRB + SGM | $\mathbf{100.0 \pm .1}\%$ | 400K | PROPRIO | POINTENV |
| SPTM | $39.3 \pm 4.0\%$ | - | VISUAL | VIZDOOM |
| SPTM + SGM | $\mathbf{60.7 \pm 4.0}\%$ | 114K | VISUAL | VIZDOOM |
| CONSPTM | $68.2 \pm 4.1\%$ | 1M | VISUAL | SAFETYGYM |
| CONSPTM + SGM | $\mathbf{92.9 \pm 1.4}\%$ | 1M | VISUAL | SAFETYGYM |

Table 1: SGM boosts performance of all existing state-of-the-art semi-parametric graphical methods.

image features, asymmetric distance metric, or low-level controller. Below, we describe our training procedure in detail.

We benchmark against the two available environments used by the SoRB and SPTM baselines, and an additional visual navigation environment. These range in complexity and are shown in Figure 5. The environment descriptions are as follows: **PointEnv**[9] continuous control of a point-mass in a maze used in SoRB. Observations and goals are positional $(x, y)$ coordinates. **ViZDoom**[49] discrete control of an agent in a visual maze environment used in SPTM. Observations and goals are images. **SafetyGym**[38] continuous control of an agent in a visual maze environment. Observations and goals are images, though odometry data is available for observations but not for goals.

We utilize the PointEnv maze for RL experiments, where SGM is constructed using undiscounted Q-functions with a sparse reward of 0 (goal-reached) and $-1$ (goal not reached). We increase the difficulty of this environment by thinning the walls in the maze, which exposes errors in the distance metric since two nearby coordinates may be on either side of a maze wall. SoRB also ran visual experiments on the SUNCG houses data set [46], but these environments are no longer public.

To evaluate SGM in image-based environments, we use the ViZDoom navigation environment and pretrained networks from SPTM. In addition, we evaluate navigation in the OpenAI SafetyGym [37]. In both environments, the graph is constructed over *visual first-person view observations* in a large space with obstacles, reused textures, and walls. Such observations pose a real challenge for learning distance metrics, since they are both high-dimensional and perceptually aliased: there are many visually similar images that are temporally far apart. We also implemented state-of-the-art RL algorithms without the graphical planner, such as DDPG [27] and SAC [16] with Hindsight Experience Replay [1], but found that these methods achieved near-zero percent success rates on all three environments, and were only able to reach nearby goals. For this reason, we did not include these baselines in our figures.

## 6 Results

**SGM increases robustness of plans:** We compare how SGM performs relative to prior neural topological methods in Table 1. **SGM sets a new state-of-the-art in terms of success rate** on all three environments tested and, on average, **outperforms prior methods by 2.1x**. In fact, thinning the walls in the PointEnv maze *breaks the policy from the SoRB baseline* because it introduces faulty edges through the walls. SoRB is not robust to these faulty edges and achieves a 28% score even after 400k steps of self-supervised cleanup. SGM, on the other hand, is able to remove faulty edges by merging according to two-way consistency and performing cleanup, robustly achieving a 100% success rate.

| TECHNIQUE | EASY ≤ 200 M | MEDIUM ≤ 400 M | HARD ≤ 600 M | OVERALL |
|---|---|---|---|---|
| RANDOM ACTIONS | 58.0% | 21.5% | 12.0% | 30.5% |
| VISUAL CONTROLLER | 75.0% | 34.5% | 18.5% | 42.7% |
| SPTM, SUBSAMPLED OBSERVATIONS | 70.0% | 34.0% | 14.0% | 39.3% |
| SPTM + SGM + 54K CLEANUP STEPS | 88.0% | 52.0% | **26.0%** | 55.3% |
| SPTM + SGM + 114K CLEANUP STEPS | **92.0%** | **64.0%** | **26.0%** | **60.7%** |

Table 2: SGM improves graph-based success rates across goal difficulties in ViZDoom. More cleanup steps yield better performance because more infeasible transitions are removed from the graph.

Similarly, in visual environments, the sparse graph induced by SGM produces significantly more robust plans than the SPTM baselines. While SGM solves the SafetyGym environment with a 96.6% success rate, it navigates to only 60.7% of goals in ViZDoom. Note that this VizDoom success rate is lower than in [41] because [41] uses human demonstrations whereas we do not. We found that the primary source of error in VizDoom was due to perceptual aliasing where two perceptually similar observations are actually far apart. For further details, see Appendix B.

We examine SGM performance with and without cleanup. Figure 6 shows that success rapidly increases as the sparse graph is corrected. However, without sparsity, the number of errors is too large to quickly correct. Success rates in Table 3 show that **sparsity-enabled cleanup is essential for robust planning** and that **cleanup improves mean performance by 2.4x**.

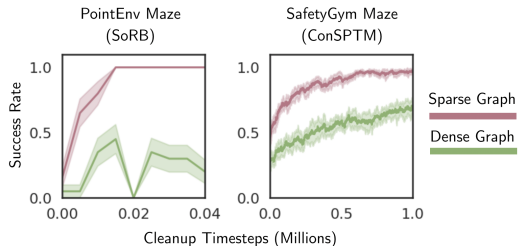

We also study success as a function of goal difficulty in Table 2, where difficulty is determined by the ground-truth distance between the initial agent's position and its goal. Even with cleanup, performance degrades with goal difficulty, which is likely the result of perceptual aliasing causing faulty edges (Appendix B).

Figure 6: Success rate as a function of cleanup steps in PointEnv (FourRooms maze) and SafetyGym. SGM is rapidly corrected while SoRB, because of errors in its dense graph, is infeasible to clean. SPTM can be cleaned, but only slowly.

**Two-way consistency is the preferred subsampling strategy:** We ablate different subsampling strategies to examine the source of our performance gains. We compare SGM with two-way consistency with variants of SGM with one-way consistency, where connectivity between two nodes is determined by only checking either incoming or outgoing distance function values, as well as a simple random subsampling strategy where a subset of nodes and edges are uniformly removed from the dense graph. Table 3 shows

| CRITERION | W/O CLEANUP | W/ CLEANUP |
|---|---|---|
| UNIFORM | 31.2 ± 3.8% | 64.6 ± 3.4% |
| PERCEPTUAL | 31.8 ± 3.6% | 77.0 ± 2.5% |
| INCOMING | 33.6 ± 3.1% | 84.1 ± 2.0% |
| OUTGOING | 28.7 ± 3.2% | 86.8 ± 2.9% |
| TWO WAY | **38.2 ± 3.8%** | **92.9 ± 1.2%** |

Table 3: Comparing node subsampling strategies in SafetyGym. Two-way consistency improves success rate relative to other criteria.

success rates for a SafetyGym navigation task before cleanup and after 1M steps of cleanup. Before cleanup, all plans perform poorly with a maximum 38.2% success rate achieved by two-way SGM. After cleanup the SGM variants significantly outperform a naive uniform subsampling strategy, and subsampling with two-way consistency achieves the highest success rate before and after cleanup.

**Investigation of two-way consistent graph quality:** We further investigate how individual components of two-way consistent SGM affect final graph and plan quality. In Figure 7 (left), we display the total edge count for various edge lengths in the ViZDoom graphical memory after different steps of the SGM algorithm. Since nodes should only be connected locally, long edges are most likely to be faulty, representing infeasible transitions. Although the different components of the SGM algorithm all help reduce errors, **two-way consistency removes the most incorrect edges** from the original dense graph while preserving the most correct edges.

Finally, in Figure 7 (right) we display the average number of steps required for different agents to reach their goal in ViZDoom. The average is only taken over successful paths, and the dotted line

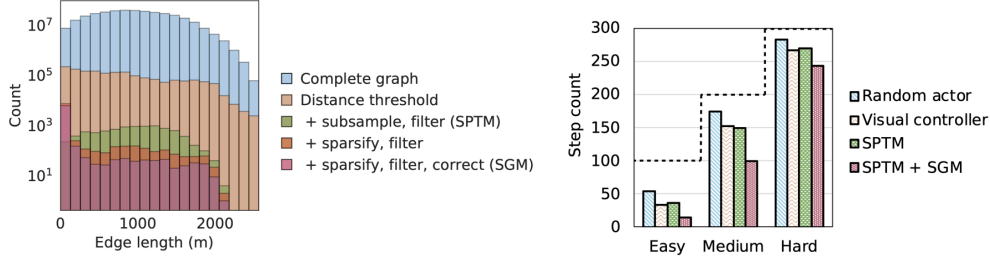

Figure 7: (Left) Overlayed histograms of edge lengths for graphical memories in ViZDoom, where long edges are incorrect. Graph quality significantly improves with SGM, which reduces the frequency of long edges. (Right) SGM reaches goals using the fewest environment steps (shortest path length).

shows the maximum number of steps allowed. We show that, on average, agents equipped with sparse graphical memory take the fewest steps to reach their goals compared to other methods. This suggests that shorter paths are another reason for improved success relative to other methods.

# 7   Conclusion

In this work, we proposed a new data structure: an efficient, sparse graphical memory that allows an agent to consolidate many environment states, model its capability to traverse between states, and correct errors. In a range of difficult visual and coordinate-based navigation environments, we demonstrate significantly higher success rates, shorter rollouts, and faster execution over dense graph baselines and learned controllers. We showed that our novel two-way consistency aggregation criterion is critical to this success, both theoretically and empirically. We hope that this direction of combining classic search-based planning with modern learning techniques will enable efficient approaches to long-horizon sensorimotor control tasks. In particular, we see scaling sparse graphical memory to challenging manipulation tasks as a key outstanding challenge for future work.

## Broader impact

**Interpretability**   To build trust in deep RL systems, users would, ideally, be able to interrogate the system: "What is the deep RL system going to do? Why?" With state-of-the-art model-free approaches such as proximal policy optimization [44], it is not possible to answer these questions – the policy network is a black box. The explicit graphical plans produced by SGM, in contrast, can provide a partial answer to these questions. The future nodes in a plan given by SGM indicate what it is attempting to do, and errors in an overall plan can be debugged by tracing them to individual faulty edges in the graph. For this reason, we see graphical planning methods such as SGM as advantageous for the trust and interpretability of deep RL systems relative to model-free methods.

**Safety**   SGM assumes that the agent's only reward signal is an indication of whether or not the current state satisfies the goal. This problem formulation ignores potential damage that could be caused during the intermediate steps taken to reach the goal, which, depending on the application, could be significant. Designing an RL agent that can safely explore and reach goals in its environment is a fundamental challenge for the entire field.

**Real-world applications**   While methods for long-horizon RL are highly general, such as the method we present in this paper, the most immediate potential applications of our method are in robotics. Completely solving the problem formulation in this paper, long-horizon control from high-dimensional input with sparse reward, would have wide-sweeping impact across robotic applications.

## Acknowledgments

This work was supported in part by Berkeley Deep Drive (BDD), ONR PECASE N000141612723, Open Philanthropy Foundation, NSF CISE Expeditions Award CCF-1730628, DARPA through the LwLL program, the DOE CSGF under grant number DE-SC0020347, the NSF GRFP under grant number DGE-1752814, and Komatsu. DP is supported by Google FRA.

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
