[Supplementary Material · 2020_neurips_graphs_appendix.pdf]

# Appendix

## A  Environments & Hyperparameters

### A.1  Perceptual consistency baseline and fast-path

Two-way consistency (1-2) involves a maximization over the replay buffer, which can be costly. We use a pairwise, symmetric *perceptual consistency* criterion as a fast test of similarity in latent space to accelerate graph construction. A new state $\hat{s}$ is only added to the memory if it fails both perceptual and two-way consistency with each state already in the memory, in which case we consider it novel and useful to retain for planning. This allows us to skip the TWC check for substantially different states.

We say that $\hat{s}$ is perceptually consistent with a previously recorded state $s \in \mathcal{V}$ if

$$||\phi(s) - \phi(\hat{s})||_2 \leq \tau_p, \tag{4}$$

where $||\phi(s) - \phi(\hat{s})||_2$ measures the visual similarity of states that the agent encounters through the $l_2$ distance between embeddings of each state. In proprioceptive tasks, the identity function is used for the embedding $\phi(\cdot)$. However, in visual navigation, nearby locations can correspond to images that are significantly different in pixel space, such as when an agent rotates [41]. To mitigate this problem, for high-dimensional images, $\phi(\cdot)$ is a learned embedding network such as a $\beta$-VAE [19, 24] for SafetyGym or a subnetwork of the distance function for ViZDoom.

Table 3 shows that two-way consistency significantly outperforms perceptual consistency on its own, so the TWC criterion is still important. Theorem 1 still holds when combining the strategies as (4) can only make aggregation more conservative.

### A.2  Low-level Controller

For the RL experiments, we use actor-critic methods to train a low-level controller (the actor) and corresponding distance metric (the critic) simultaneously. In particular, we use distributional RL and D3PG, a variant of deep deterministic policy gradient [3, 27]. For experiments on SafetyGym, we use a proprioceptive state-based controller for both SGM and the dense baseline. For the SSL experiments in ViZDoom, we use the trained, behavior cloned visual controller from SPTM. The controller is trained to predict actions from a dataset of random rollouts, where goals are given by achieved visual states.

### A.3  Environments

In our experiments, we tune SGM hyperparameters to maintain graph connectivity and achieve a desired sparsity level. Thresholds are environment-specific due to different scaling of the distance function. Success rate is only evaluated after selecting parameters.

**PointEnv:** PointEnv is maze environment introduced in [9] where the observation space is proprioceptive. We run all SoRB experiments in this environment. The episodic return is undiscounted $\gamma = 1$, and the reward is an indicator function: $r = 0$ if the agent reaches its goal and $r = -1$ otherwise. The distance to goals can thus be approximated as $d = |Q(s,a)|$. We approximate a distributional $Q$ function, which serves as a critic, with a neural network that first processes the observation with a 256-unit, fully-connected layer, that then merges this processed observation with the action, and that then passes the observation-action combination through another 256-unit, fully-connected layer. For an actor, we use a fully-connected network that has two layers of 256 units each. Throughout, we use ReLU activations and train with an Adam optimizer [23] with a step size of 0.0003. To evaluate distances, we use an ensemble of three such distributional $Q$ functions, and we pessimistically aggregate across the ensemble. For SGM, we set MAXDIST $= 10$, $\tau_p = 0.05$, $\tau_a = 5$, $k = 5$, MAXSTEPS $= 30$, and ACTINGCUTOFF $= 1$ for the localization threshold. For SoRB, we set MAXDIST $= 6$, $k = 5$ and MAXSTEPS $= 18$ due to a higher density of states.

**ViZDoom:** For our ViZDoom visual maze navigation experiments, we use the large training maze environment of [41]. Following [41], the distance metric is a binary classifer trained with a Siamese network using a ResNet-18 architecture. The convolutional encoder embeds image observations into a 512 dimensional latent vector. Two image embeddings are then concatenated and passed

Figure 8: Observations passing perceptual (feature difference threshold) and two-way consistency criteria for SafetyGym using a $\beta$-VAE for perceptual features and a contrastive distance for two-way consistency. The two-way consistent observations are, unsurprisingly, more diverse in orientation than perceptually nearby ones. Although the majority of two-way consistent observations are correct *i.e.* nearby in space, there is one false positive (blue floor).

Figure 9: Construction of Sparse Graphical Memory in the ViZDoom environment. We add nodes from a source replay buffer that unevenly covers the environment (left), creating a sparsified memory. k-nearest neighbor edge filtration limits the number of errors, which are further corrected via cleanup. The image memory in SGM much more evenly covers the environment, even though no coordinate information is used during graph construction.

through a 4 layer dense network with ReLU activations, 512 hidden units, and a binary cross entropy objective where $y = 1$ if the two embeddings are temporally close and $y = 0$ otherwise. An Adam optimizer [23] with a step size of $0.0001$ is used for gradient updates to finetune the pretrained network of [41]. For the controller, we use the pretrained network of [41] with no finetuning.

For graph creation, we collect a replay buffer of 100 episodes of random actions, each consisting of 200 steps (*i.e.* 20,100 total images) and add each image sequentially to SGM to simulate an online data collection process. For both SPTM and SPTM with SGM, we set MAXDIST = 2, $k = 5$, ACTINGCUTOFF = 5.75, and MAXSTEPS = 10. For SGM, $\tau_p = 20$ and $\tau_a = 2$. To make the SPTM baseline tractable, we randomly subsample the replay buffer to 2,087 states (the same size as our sparsified vertex set), as edge creation is $O(|\mathcal{V}|^2)$ and start/goal node localization is $O(|\mathcal{V}|)$. The baseline graph has $2,087$ nodes and $18,921$ edges. While we can evaluate the baseline with a graph consisting of all $20,100$ states, this is a dense oracle that has $20,100$ nodes and $1,734,524$ edges and takes hours to construct. The oracle achieves $75\%, 55\%$, and $35\%$ success rates at easy, medium and hard goal difficulties ($55.0\% \pm 6.4\%$ overall).

To increase the robustness of edge creation under perceptual aliasing, we aggregate the distance over temporal windows in the random exploration sequence. For states $s_t^{(i)}$ in episode $i$ and $s_{t'}^{(j)}$ in episode $j$, we set the distance to the maximum pairwise distance between states $s_{t-2}^{(i)}, s_{t-1}^{(i)}, s_t^{(i)}, s_{t+1}^{(i)}, s_{t+2}^{(i)}$ and

Figure 10: A visualization of the fine-tuned SPTM distance metric in the ViZDoom. The agent sees the reference image on left. The second image shows the acting distance between the reference image and states previously seen throughout the maze according to $d(s_{\text{agent}}, \cdot)$. A coordinate is colored yellow if the associated state is close to the reference in acting distance, while green and blue coordinates are distant with respect to the reference state. Aggregating the distance pessimistically across temporal windows (third image) reduces false positives that are distant in coordinate space but close in acting distance space. In the fourth image, we threshold the aggregated distance according to $\tau_a$. While most states passing the threshold are near the agent, some are distant. These are false positives. In the rightmost figure, we show histograms of the aggregated acting distance for negative, distant pairs of states (top) and close, positive pairs (bottom), totaling 404M pairs.

states $s^{(j)}_{t'-2}, s^{(j)}_{t'-1}, s^{(j)}_{t'}, s^{(j)}_{t'+1}, s^{(j)}_{t'+2}$, aggregating over up to 25 pairs. In contrast, SPTM aggregated with the median and compared only 5 pairs. Our aggregation is more pessimistic as our replay buffer is created with random exploration that suffers from extensive perceptual aliasing rather than a human demonstrator that mostly stays in the center of hallways. Figure 9 shows the graph construction process.

**SafetyGym:** In the SafetyGym environment we employ a contrastive objective to discriminate between observations from the same window of a rollout and random samples from the replay buffer. The contrastive objective is a multiclass cross entropy over logits defined by a bilinear inner product of the form $f(z_t, z_{t'}) = z_t^T W z_{t'}$, where $W$ is a parameter matrix, and the distance scores are probabilities $d = \exp(-f(z_t, z_{t'}))$. To embed the observations, which are $64 \times 64$ rgb images, we use a 3 layer convolutional network with ReLU activations with a dense layer followed by a LayerNorm to flatten the output to a latent dimension of 50 units. We then train the square matrix $W$ to optimize the ccontrastive energy function. As before, we use Adam [23] with a step size of $0.0001$ for optimization.

The $\beta$-VAE maximum likelihood generative model for perceptual features has an identical architecture to the distance metric but without the square matrix $W$. Each image is transformed into an embedding, which is stored in the node of the graph. When a new image is seen, to isolate visually similar neighbors, we compute the L2 distance between the latent embedding of the image and all other nodes in the graph. Since computing the L2 distance is a simple vector operation, it is much more computationally efficient than querying the distance function, which requires a forward pass through a neural network, $O(|\mathcal{V}|)$ times at each timestep.

Perceptually consistent and two-way consistent observations relative to an agent's current observation are shown in Figure 8. For constructing both dense and sparse graphs, we use MAXSTEPS $= 9$, $k = 6$, $\tau_p = 6$, and $\tau_a = 5$.

## B  Perceptual Aliasing with Learned Distance

A common issue with learning distances from images is perceptual aliasing, which occurs when two images are visually similar but were collected far apart in the environment. We examine a heatmap of learned distances in ViZDoom in Figure 10. Although most observations with small distance are correctly clustered around the agent's location, there are several clusters of false positive observations throughout the map due to perceptual aliasing between parts of the maze. Perceptual aliasing results in wormhole connections throughout the graph where two distant nodes are connected by an edge, which creates an attractor for path planning. We show an example path planned by an agent that is corrupted by perceptual aliasing in Figure 11. In its plan, the agent draws a connection between two visually identical but distant walls, which corrupts its entire plan to reach the goal.

Figure 11: Failure mode in ViZDoom planning when cleanup and pessimistic distance aggregation are not used. While waypoints in the plan between start and goal states are closely grouped for much of the path, the planner exploits the perceptual aliasing of walls in the environment as a shortcut through the environment. Pessimistic aggregation of the distance metric can help with the issue, but does not fully resolve the problem. By stepping through the environment during cleanup, we can remove the remaining untraversable edges.

False positives can be reduced further by aggregating the distance pessimistically across consecutively collected observations. However, doing so does not eliminate them altogether. The presence of false positives further supports the argument for sparsity. With sparsity and cleanup, it is possible to remove the majority of incorrect edges to yield robust plans.

## C    Re-planning with a Sparse Graph

Figure 12 shows an example of an evaluation rollout, which includes a cleanup step when the agent encounters an unreachable waypoint. The agent creates an initial plan and moves along the proposed waypoints until it encounters an obstacle. Unable to pass the wall (represented by the blue blocks), the agent removes the edge between two nodes across the wall and re-plans. Its second plan has no obstacles and it is therefore able to reach its goal.

## D    Proof Bounding the Cost Gap of Two-way Consistency

**Theorem 1.** *Let $\mathcal{G}_\mathcal{B}$ be a graph with all states from a replay buffer $\mathcal{B}$ and weights from a distance function $d(\cdot, \cdot)$. Suppose $\mathcal{G}_{TWC}$ is a graph formed by aggregating nodes in $\mathcal{G}_\mathcal{B}$ according to two-way consistency $C_{out}$ and $C_{in} \leq \tau_\alpha$. For any shortest path $P_{TWC}$ in $G_{TWC}$, consider the corresponding shortest path $P_\mathcal{B}$ in $G_\mathcal{B}$ that connects the same start and goal nodes. Suppose $\mathcal{P}_\mathcal{B}$ has $k$ edges. Then:*

  *(i) The weighted path length of $P_{TWC}$ minus the weighted path length of $P_\mathcal{B}$ is no more than $2k\tau_\alpha$.*

Figure 12: Evaluation of SGM in SafetyGym. This figure is a top-down view abstraction of the SafetyGym environment made to cleanly represent the sparse graph. The actual environment is more visually complex and the agent sees first-person view images.

*(ii) Furthermore, if $d(\cdot, \cdot)$ has error at most $\epsilon$, the weighted path length of $P_{TWC}$ is within $k\epsilon + 2k\tau_\alpha$ the true weighted distance along $\mathcal{P}_\mathcal{B}$.*

*Proof.* Before proceeding with the proof, we establish more formal notation about the statement of the theorem.

We denote the replay buffer graph $\mathcal{G}_\mathcal{B} = (\mathcal{V}_\mathcal{B}, \mathcal{E}_\mathcal{B}, \mathcal{W}_\mathcal{B})$. Aggregating nodes according to TWC leads to the subgraph $\mathcal{G}_{TWC} = (\mathcal{V}_{TWC}, \mathcal{E}_{TWC}, \mathcal{W}_{TWC})$ where all $V_\mathcal{B} \backslash V_{TWC}$ satisfy two-way consistency at threshold $\tau_a$ with some node in $V_{TWC}$.

For any start and goal node $s_1, s_{k+1} \in V_{TWC}$, consider the shortest path that connects them in $G_\mathcal{B}$: $P_\mathcal{B} = (s_1, s_2, \ldots, s_{k+1})$. Suppose the edge weights of $P_\mathcal{B}$ are $(w_1, w_2, \ldots, w_k)$, given by the distance function according to $w_1 = d(s_1, s_2)$, etc.

Proof of (i): We will show that there exists a corresponding shortest path in $\mathcal{G}_{TWC}$ given by $P_{TWC} = (s_1, s_2', \ldots, s_k', s_{k+1})$ with edge weights (distances) $(w_1', w_2', \ldots, w_k')$ where the total weight of $P'$ is no more than $2k\tau_a$ the total weight of $P_\mathcal{B}$, i.e., $\sum_i w_i' \leq 2k\tau_a + \sum_i w_i$.

We proceed by construction by first specifying $s_i'$ for $i \in \{2, 3, \ldots, k\}$. If $s_i \in \mathcal{V}_{TWC}$, let $s_i' = s_i$. Otherwise, let $s_i'$ be any node in $\mathcal{V}_{TWC}$ satisfying two-way consistency with $s_i$.

Given this choice of $P_{TWC}$, we will show for arbitrary $j \in \{1, 2, \ldots, k\}$ that

$$w_j' = d(s_j', s_{j+1}') \leq 2\tau_a + d(s_j, s_{j+1}) = 2\tau_a + w_j.$$

By outgoing two-way consistency between $s_j$ and $s_j'$ (or because $s_j = s_j'$), we have

$$|d(s_j, s_{j+1}) - d(s_j', s_{j+1})| \leq \tau_a.$$

Similarly, by incoming two-way consistency between $s_{j+1}$ and $s_{j+1}'$ (or because $s_{j+1} = s_{j+1}'$),

$$|d(s_j', s_{j+1}) - d(s_j', s_{j+1}')| \leq \tau_a.$$

By the triangle inequality, $|d(s_j, s_{j+1}) - d(s_j', s_{j+1}')| \leq 2\tau_a$, i.e., $w_j' \leq w_j + 2\tau_a$. Finally, we note that the result (i) follows by summing the bound $w_j' \leq 2\tau_a + w_j$ over all indices $j$.

Proof of (ii): Furthermore, assume that the edge weights of $P_\mathcal{B}$ given by $(w_1, w_2, \ldots, w_k)$ all have error at most $\epsilon$. That is to say, if $(w_1^*, w_2^*, \ldots, w_k^*)$ denote the true distances along the path $P_\mathcal{B}$, assume $|w_i - w_i^*| \leq \epsilon$ for all $i \in \{1, 2, \ldots, k\}$. We will show that the shortest weighted path length in the aggregated graph $\mathcal{G}_{TWC}$ differs from the true weighted length of the shortest path in $\mathcal{G}_\mathcal{B}$ by at most $k\epsilon + 2k\tau_\alpha$, *i.e.*, $|\sum_i w_i' - \sum_i w_i^*| \leq k\epsilon + 2k\tau_\alpha$.

In the proof of (i), we showed $\sum_i w_i' \leq 2k\tau_a + \sum_i w_i$. By the additional assumption of part (ii), we have $w_i \leq w_i^* + \epsilon$ for all $i \in \{1, 2, \ldots, k\}$. Chaining these two inequalities yields $\sum_i w_i' \leq k\epsilon + 2k\tau_a + \sum_i w_i^*$.

Similarly, the proof of (i) shows $|d(s_j, s_{j+1}) - d(s_j', s_{j+1}')| \leq 2\tau_a$ for arbitrary $j \in \{1, 2, \ldots, k\}$. Hence, $w_j \leq w_j' + 2\tau_\alpha$. Rearranging and summing this bound yields $-2k\tau_\alpha + \sum_j w_j \leq \sum_j w_j'$. Moreover, as for part (ii) we assume $|w_i - w_i^*| \leq \epsilon$ for all $i \in \{1, 2, \ldots, k\}$, we also have $w_j^* - \epsilon \leq w_j$. Summing and chaining inequalities yields $-k\epsilon - 2k\tau_\alpha + \sum_i w_i^* \leq \sum_i w_i'$.

Together, the results of the previous two paragraphs yield what is desired: $|\sum_i w_i' - \sum_i w_i^*| \leq k\epsilon + 2k\tau_\alpha$. $\qquad\qquad\square$

# E    Planning speed of SGM

Table 4 shows the time required to take a single action including graph planning on a dense graph constructed using SoRB and with a sparse graph in the PointEnv environment. Sparse Graphical Memory significantly accelerates action selection.

| METHOD | TIME TO TAKE ACTION (S) |
|---|---|
| SoRB | $0.550 \pm 0.220$ |
| SGM (OURS) | $\mathbf{0.077 \pm 0.004}$ |

Table 4: The average and standard-deviation wall-clock time for taking an action with SGM (our method) and SoRB (previous state-of-the-art) in PointEnv.

# F    Reproducibility checklist

**Models & Algorithms**    Section 5 and Section A describe our model architectures, which follow past work in the two previously studied planning environments (PointEnv and ViZDoom). We analyze the complexity of graph construction with our online approximation in Section 4.2 and of edge creation and agent localization in Section A.3.

**Theoretical Claims**    We state and prove Theorem 1, which (i) bounds the increase in plan cost when aggregating states via two-way consistency and (ii) given an existing error bound on the distance function, bounds the additional error induced by aggregating states via two-way consistency. The theorem is stated informally in the main paper and precisely with a proof in Section D.

**Datasets**    Environments and experience collection are described in Section A.

**Code**    Code and documentation are included on the project website: https://mishalaskin.github.io/sgm/.

**Experimental Results**    We describe the setup of our experiments in Section 5 and Section A, including hyperparameter selection and specification. Runtimes are described, including in Section E.