[Reviews · NeurIPS 2020]

Review 1

Summary and Contributions: This paper proposed a new hierarchical RL method with two-level controllers(a high-level controller and a low-level controller). They build a Sparse Graphical Memory(SGM) to obtain an abstract(smaller) MDP from the original MDP, which reduces the difficulty of the planning problem. The high-level controller plans on the SGM and produces sub-goals(waypoints) for the low-level controller. The main contribution is that they propose a novel node-merging method called two-way consistency(TWC). The experiments show that SGM with TWC and memory cleanup can increase success rates of navigation tasks. I have read the reponse, and I will keep the same score on this paper.

Strengths: The method(SGM+TWC) proposed in this paper is reasonable and the author also gives out the proof for their theorem. The experimental results are good and the ablation study shows some of the properties of their method.

Weaknesses: 1.It seems that they only evaluate their algorithm on a single map of ViZDoom and SafetyGym. Lacking of abundant experimental evaluations makes their method less convincing. 2.When memory graphs or tasks gets more complex, it's impossible to use an optimal search method to plan on the graph. The proposed algorithm may have its limitation when extended to more challenging tasks.

Correctness: The claims and method are correct. The empirical methodology is correct.

Clarity: The paper is well written and easy to understand.

Relation to Prior Work: Yes.

Reproducibility: Yes

Additional Feedback: 1. Evaluate on more maps. 2. Incroporate more powerful high-level planners or construct a multi-level(instead of two-level) control paradigm.


Review 2

Summary and Contributions: The paper proposes to solve long horizon goal-conditioned visual-navigation tasks by constructing a sparse graph of observations from the replay buffer. The key contribution is a two-way consistency criterion to aggregate similar nodes in the graph. The resulting sparse graph is shown to be more reliable and amenable to correction at execution time.

Strengths: The paper studies the important approach of combining existing reinforcement learning approaches with more classical planning elements so as to solve long-horizon problems with variable goals. The basic idea of abstracting a sparse graph from the observations seems sound. The paper attaches code to aid reproducibility.

Weaknesses: The main novelty of the paper is an approach to aggregate related observations together to construct a sparse topological map; many previous works have also explored this idea, such as [1, 2, 3], which are not addressed in the paper. The paper makes no distinction between state and observations. The fact that high-level actions are also observations makes the presentation confusing. Furthermore, this limits the class of tasks that can be addressed. The main theorem in the paper make strong assumptions on the Q function (see correctness) The comparison to SPTM is unclear. The results reported in the original SPTM paper are much better than the ones reported here. Is it because of the subsampled observations? If so, why is this a valid comparison? Also, in SPTM, some care was taken to ensure that the approach generalized to unseen environments; there was a training environment, some validation environments and test environments. In this paper, there does not appear to be any attempt to do this. The appendix describes 3 different perceptual models used for perceptual consistency as three different lower level policies trained for each separate environment. The importance of this perceptual similarity is not addressed in the body of the paper. The Q function is typically smaller than models used for perceptual embeddings, so it seems unlikely that its faster computationally to compute a perceptual embedding. Presumably directly using Q function similarity leads to poor performance and so this environment specific embedding was required. There are a substantial number of hyperparameters tuned for each environment, which raises questions as to the generality of the approach. Not all results have confidence intervals included. Details of seeds for different runs are not described. [1] Neural Topological SLAM for Visual Navigation, Devendra Singh Chaplot, Ruslan Salakhutdinov, Abhinav Gupta, Saurabh Gupta, CVPR 2020 [2] A Behavioral Approach to Visual Navigation with Graph Localization Networks Kevin Chen, Juan Pablo de Vicente, Gabriel Sepulveda, Fei Xia, Alvaro Soto, Marynel Vazquez, Silvio Savarese, RSS 2019 [3] Scaling Local Control to Large-Scale Topological Navigation Xiangyun Meng, Nathan Ratliff, Yu Xiang, Dieter Fox, ICRA 2019

Correctness: The central theorem of the paper, Theorem 1, appears to be correct, but does not quite validate that “goals can be reached with almost the same number of steps in both the aggregated and dense graphs when TWC is used, i.e. the resulting plans are near-optimal” [L71]. The theorem shows that plans are near optimal if we have an accurate Q function – in which case you can already accomplish the task perfectly. Instead, a theorem showing optimality bounds of the proposed approach, given an existing optimality bound on Q would be more useful.

Clarity: The paper is generally easy to read but too many of the details, in particular, with respect to perceptual similarity are left to the appendix. In general, I was confused as to precisely how the testing was carried out.

Relation to Prior Work: The paper is missing comparisons with a number of recent papers exploring topological memory (see weaknesses). The paper does not discuss differences between the proposed approach with the listed papers.

Reproducibility: Yes

Additional Feedback: -- Post Rebuttal -- Thank you for the clarifications.


Review 3

Summary and Contributions: This paper attempts to combine the strengths of deep learning and classical planning methods in a scalable manner, to solve long-horizon visual navigation tasks. The authors introduce Sparse Graphical Memory (SGM), an algorithm to sparsify a graph consisting of observations of an agent as nodes and the transitions as edges. The key idea in the algorithm is that two nodes/observations are aggregated if they are both interchangeable as goals or starting states. The authors demonstrate the effectiveness of SGM on three different environments, to outperform two baseline methods.

Strengths: Previous works such as SoRB and SPTM build a graph of replay buffer with observations as nodes and transitions as edges and use graph-based planning to find waypoints between the start and goal states and low-level neural controllers to traverse between waypoints. This approach is not scalable to large environments as the graphs grow quadratically. This paper introduces an algorithm to prune the nodes in this graph to ensure scalability and simple heuristics to cleanup erroneous transitions. The pruning algorithm is derived by extending the notion of value irrelevance to goal-conditioned RL: two nodes are redundant if they are both interchangeable as goals and start states, according to the goal-conditioned value function. The authors perform ablation studies to demonstrate that the improved performance is due to the effectiveness of the proposed method.

Weaknesses: - Although it is remarked that long-horizon visual navigation tasks are challenging for MPC, policy optimization and Q-learning algorithms, SGM is not compared with such methods on these tasks. For example, Hindsight Experience Replay. - Completely different network architectures and hyperparameters are used for each task.

Correctness: The empirical methodology is same as that of previous works. Use of different settings for each benchmark task is concerning.

Clarity: The paper is well-written and easy to read.

Relation to Prior Work: Relation to prior work is clearly discussed in Section 2.

Reproducibility: Yes

Additional Feedback: - SPTM originally uses human demonstrations to bootstrap the graph. How does that influence the results presented in this paper? - The paper does not work with some PDF readers and page 13 in the Appendix is missing. Update after author response: I acknowledge that I have read the rebuttal.

[Author Response · NeurIPS 2020]

We thank the reviewers for their feedback. We are glad that they found the problem of combining classical planning with Reinforcement Learning important (R3), our experimental results and ablations to be compelling (R1,R5), and our method to be sound (R1,R3). We are also pleased that the reviewers clearly identified the main contribution of our work – a general procedure for automatically pruning nodes in a graph over observations called Two-Way Consistency (TWC) (R1,R3,R5). We address the reviewers' feedback below and will incorporate all of it.

———————————————————— **Shared Feedback (R1,R3,R5)** ————————————————————

**Comparison with SPTM** (R3,R5) *"[is] original SPTM ... better ... because of the subsampled observations?"* (R3) *"SPTM originally uses human demonstrations"* (R5) The performance difference is not due to subsampling, but because the original SPTM paper uses human demonstrations. We follow the more general and more difficult problem statement of SoRB: long-horizon planning without demonstrations.

**More test-time maps** (R1,R3) *"evaluate ... on a single map of ViZDoom and SafetyGym."* (R1) *"In SPTM... there was a training ... validation ... and test environments."* (R3). We used SoRB's problem statement: an exploration phase followed by a deployment phase in the same maze. SPTM can only generalize to new mazes with an expert walkthrough of the new maze, whereas we study goal generalization in the same maze by sampling goals and starting points randomly with no demonstrations (exactly as in SoRB). Furthermore, we test in three diverse environments (PointEnv, VizDoom, SafetyGym), which is more than SoRB (two envs) and SPTM (one env).

**Hyperparameters** (R3,R5) *"Substantial number of hyperparameters [are] tuned for each environment..."* (R3) We used the exact architecture / hyperparameters of SoRB & SPTM. We only tune the thresholds $\tau_\alpha$ (TWC) & $\tau_p$ (perceptual).

———————————————————— **For Reviewer #1 (R1)** ————————————————————

**Complex graphs**. *"It's impossible to use an optimal search method to plan on [more complex] graphs. The proposed algorithm may have its limitation when extended to more challenging tasks."* In PointEnv, we found that only 0.0038% of SGM's time to choose an action was taken up by graph search. We expect the benefits of SGM will only increase as graphs become more complex. SGM uses Dijkstra's algorithm, with $O(|\mathcal{E}|\log|\mathcal{V}|)$ complexity. With Line 12 of Alg 1 (k-NN edge filtering), $|\mathcal{E}|$ is a constant, and we can control the size of $|\mathcal{V}|$ by tuning $\tau_\alpha$.

———————————————————— **For Reviewer #3 (R3)** ————————————————————

**Assumptions of theorem**. *"The main theorem in the paper makes strong assumptions on the Q function."* Our main theorem makes no assumptions on the Q function. Instead, it bounds the additional error when using TWC to turn a dense graph into a sparse graph. The bound holds regardless of the original error in the dense graph.

**Better optimality bound**. *"[A] theorem showing optimality bounds of [TWC]... given an existing optimality bound on Q would be more useful."* Thank you for the excellent suggestion. We took your feedback and proved that, given an existing optimality bound on Q with error $\epsilon$, TWC on plans of path length $k$ has error at most $k\epsilon + 2k\tau_\alpha$.

**Perceptual similarity**. *"The importance of ... perceptual similarity is not addressed... it seems unlikely that its faster computationally to compute a perceptual embedding"* Perceptual consistency is substantially faster as it computes $|\mathcal{V}|$ embeddings and their pairwise L2 distances (a cheap vectorized computation) whereas TWC requires $|\mathcal{V}|^2$ queries to the neural distance function. In Table 3, we run an ablation with perceptual consistency. Perceptual consistency alone achieves 77.0% success whereas the full method achieves 92.9% success.

**States vs observations**. *"The paper makes no distinction between state and observations."* We'll be sure to clarify. We demonstrate SGM in environments with access to state as well as with access to visual observations only. *"The fact that high-level actions are also observations... limits the class of tasks that can be addressed."* SGM is no more limited than prior graphical memory work. An observation can precisely specify a waypoint or goal as long as it includes features important for the task. Even if the observation is more specific than desired, SGM can solve many tasks because the distance function $d(\cdot,\cdot)$ can be changed, e.g. to identify such features and ignore task-irrelevant details (L185-187).

**Related work**. *"previous works... [1, 2, 3]... are not addressed in the paper."* Thank you, we will add these to the related work! Ref. 1 assumes access to a 360°camera and a pose sensor whereas we do not. In Ref. 2, the nodes in the graph are manually specified by humans whereas we automatically abstract nodes from the data. Ref. 3 has no theoretical guarantees, requires trajectories rather than unordered observations, and uses human demonstrations.

**Text suggestions**. *"Not all results have confidence intervals included. Details of seeds for different runs are not described."* Thank you for the feedback. We will add the missing confidence intervals and seed info to the text.

———————————————————— **For Reviewer #5 (R5)** ————————————————————

**Other methods**. *"SGM is not compared with... MPC, policy optimization and Q-learning"*. We found that these methods achieved near 0% success rate (a finding known from SoRB). We'll clarify this in the text.

[Meta-Review · NeurIPS 2020]

Two referees support accept, one indicates reject. Rebuttal disputes R3's point about the usefulness of the central theorem and clarifies the comparison to the baselines. I find that the additional information provided in the rebuttal overcomes most of R3's concerns and recommend acceptance of the paper. However, I would like the authors to add their new theoretical insights to the paper, clarify performance differences to those presented in the original papers on the baselines and clarify states vs observations to make the paper more readable.